# Force perceptual bias caused by muscle activity in unimanual steering

Yusuke Kishishita[1,3]*, Yoshihiro Tanaka[2], Yuichi Kurita[1,4]

**1** Hiroshima University, 1-4-1, Kagamiyama, Higashi-Hiroshima, Hiroshima, Japan, **2** Nagoya Institute of Technology, Gokiso-cho, Showa-ku, Nagoya, Aichi, Japan, **3** JSPS Research Fellow, JSPS, Tokyo, Japan, **4** JST, PRESTO, Hiroshima, Japan

* yusukekishishita@hiroshima-u.ac.jp

**Data Availability Statement:** All experimental data zip file are available from the figshare (DOI: 10.6084/m9.figshare.9791543)

**Funding:** This work was supported by JSPS KAKENHI Grant Number JP17J06986 and JST, PRESTO Grant Number JPMJPR16D3.

## Abstract

This study sought to investigate whether force perceptual bias was affected by differences in posture while steering an automobile using a psychophysical experiment to examine the relationship with muscle activity. The human perceptual characteristics of weight and force are known to be nonlinear, and a perceptual bias can occur, that is, bias that causes a perception of something that is larger or smaller than the actual scale. This is considered to be caused by physical and/or psychological conditions. Sense of effort is believed to be one influential factor. It is known to correlate with muscle activity intensity, and bias may be caused by muscle activity changes. In the current study, we hypothesized that force perceptual bias would depend on posture due to the intensity of muscle activity changes caused by changing postures during steering operation. By investigating this hypothesis, we can clarify the relationship between sense of effort and muscle activity. To investigate this issue, we conducted a psychophysical experiment to confirm postural dependence, and estimated muscle activity using a three-dimensional musculoskeletal model simulation with postural and arm force data during the experiment. In addition, prediction of bias was conducted based on a simulation in the psychophysical experiment using these data. The results revealed that bias existed, as measured by differences in postures. Additionally, a significant moderate correlation was found between the predicted bias and the actual bias, indicating the existence of a relationship between muscle activity and bias.

## Introduction

The accurate performance of human movement involves the ability to sense force/heaviness. Human perceptual characteristics are known to be nonlinear; that is, there are differences between actual force and perceived force [1, 2]. Perceived force/heaviness has traditionally been believed to depend on physical (e.g., colors, and surface condition of lifted objects) and/ or psychological (e.g., fatigue of muscle) factors, as reported by Jones et al. [3]. De Camp [4] demonstrated that perceived weight is affected by object's color, reporting that darker-colored

**Competing interests:** The authors have declared that no competing interests exist.

objects are perceived to weigh less than lighter-colored objects. Additionally, it is well known that fatigue affects sense of force/heaviness [5–7].

In daily life, an automobile is an example of a system involving a human-machine interaction based on sense of force. Sense of force is thought to be important when driving an automobile, and perceived force changes while driving. Newberry et al. [8] found that the sensation of the force exerted by the steering wheel increases with a power of 1.39, according to Stevens' power law [9] for steering wheel reaction forces ranging from 5.25–21 N and power of 0.93 for a steering wheel angles ranging from 4–16˚. These parameters of power represent the ratio of the intensity of the subject's perceived exertion of force to the actual exertion. Takemura et al. [10] investigated the perceived force characteristics for a wide range of steering angles using psychophysical experiments and reported that the characteristics followed Weber-Fechner's law [11]. This law states that perceptual intensity is proportional to the logarithm of the stimulus. It has also been reported that muscle activity changes according to the steering posture of the automobile, which changes sense of force [12].

To investigate perceptual bias, which creates a perception that is larger or smaller than the actual scale, psychophysical experiments have been conducted. Using a psychophysical experiment, van Polanen et al. [13] revealed that bias that overestimates actual weight occurs when there is a visual delay in lifting an object in a virtual reality environment. They investigated the multisensory effect (lifting an object with a visual delay) on the perceived weight [13]. Flanagan et al. [14] found that when lifting an object using a precision grip with the distal pads of the thumb and index finger, bias changed depending on the object's surface texture. When the surface texture of the lifted object is smooth, the perceived weight increase. Additionally, Sakajiri et al. [15] report that perceptual bias is generated by a difference in the reaction force direction while steering an automobile. Flanagan et al. and Sakajiri et al. report that regarding sense of effort, bias can be affected by whether muscle force functionally acts on movement. This indicates that muscle is one key factor of force/weight perception.

It has previously been reported that sense of effort and perception of force/heaviness are linked because during muscle fatigue or paralysis, we perceive both a sense of increased force/heaviness and an increase in effort [16–19]. Sense of effort is a motor command generated by the central nervous system, and it refers to a signal sent from the brain to a peripheral system. The larger the motor command, the more power a human can exert, and the size of the motor command relates to the sense of effort size. Cafarelli et al. [20] used the intensity of muscle activity as a sense of effort to investigate the relationship between muscle length and sense of force. Moreover, Morree et al. [21] provide neurophysiological evidence that movement-related cortical potential amplitude is correlated with sense of effort. Thus, previous studies have indicated that sense of force/heaviness can be evaluated based on muscle activity, which can be interpreted as sense of effort. The findings described above suggest that bias could potentially be caused by changes in muscle activity with changing postures. However, no previous studies have investigated changes in force perceptual bias caused by changes in postures. It may be possible to explain the generation mechanisms of postural dependence of force perceptual bias by comparing muscle activity intensity, which reflects sense of effort. To investigate this issue, we conducted a psychophysical experiment to confirm postural dependence, and estimated muscle activity using a three-dimensional musculoskeletal model simulation with postural and arm force data during the experiment. Additionally, the prediction of bias was carried out by the simulation in the psychophysical experiment using these data. Overall, in this study, we attempt to clarify human force discrimination in the experiment based on differences in muscle activity.

## Materials and methods

### Participants

The participants included nine healthy, right-handed subjects (nine males; mean [SD]: 21.8 [1.6] years old; 1.75 [0.06] m; 69.5 [6.7] kg). Of the nine participants, eight have official driver's licenses, and two drove on a regular basis. All participants gave written informed consent before participating in the study. Participants were paid for their time. The experimental procedures were previously approved by the local research ethics committee (Nagoya Institute of Technology).

### Apparatus

We used a simulated steering device in the experiment, as shown in Fig 1. It was the system used in [15]. A six-axis force sensor (BL Autotec, Ltd., Micro 5/50-S09) was attached at the base of each handgrip to obtain the exerted force from the hand, and the torque presentation was generated by two servomotors (maxon motor, RE40) attached to one end of the main driveshaft. Each servomotor was attached to a 21:1 reduction gear (Harmonic Drive Systems Inc., HPG-14A-21) and a rotary encoder (Microtech Laboratory Inc., ME-20) in order to apply the desired reaction force and obtain the angle. The curved handgrips were made of acrylic plastic and formed two arcs of a circle 350 millimeters in diameter.

### Procedure

Psychophysical experiments were performed using the staircase method, which included downward step and upward step, in which the test stimuli deviates from the reference stimulus (very large and very small, respectively). In this case, very large/small means the subjects could definitely perceive the difference from the reference stimulus. These test stimuli were confirmed before the experiment. The subjects were asked to compare the magnitude of the reaction force in the reference posture and the experimental posture. They grasped the handgrip with the right hand only. Each experimental posture is shown in Fig 2. The reference posture was the initial position of the steering (0˚), and the experimental postures were static postures of 30˚, 60˚, −30˚ and −60˚ from the reference. The reference stimulus was 2.0 Nm, and the

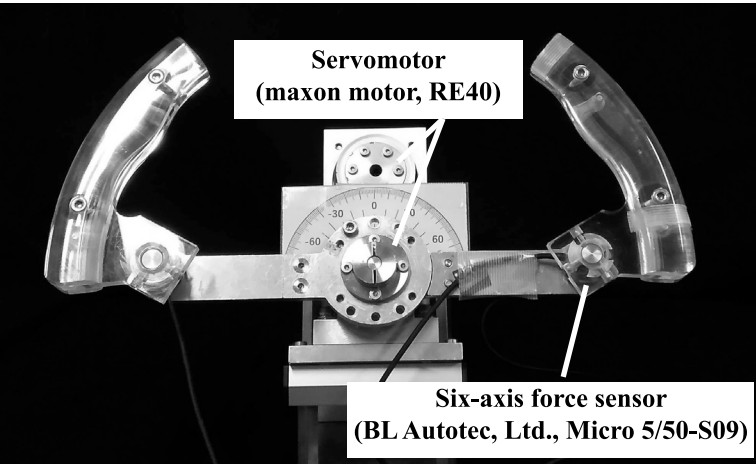

**Fig 1. The simulated steering wheel device.** The device included two servo motors to generate steering wheel torque and a six-axis force sensor was attached at the base of each handgrip to obtain the exerted force from the hand. However, we only used the right side grip in this experiment. It was the system used in [15].

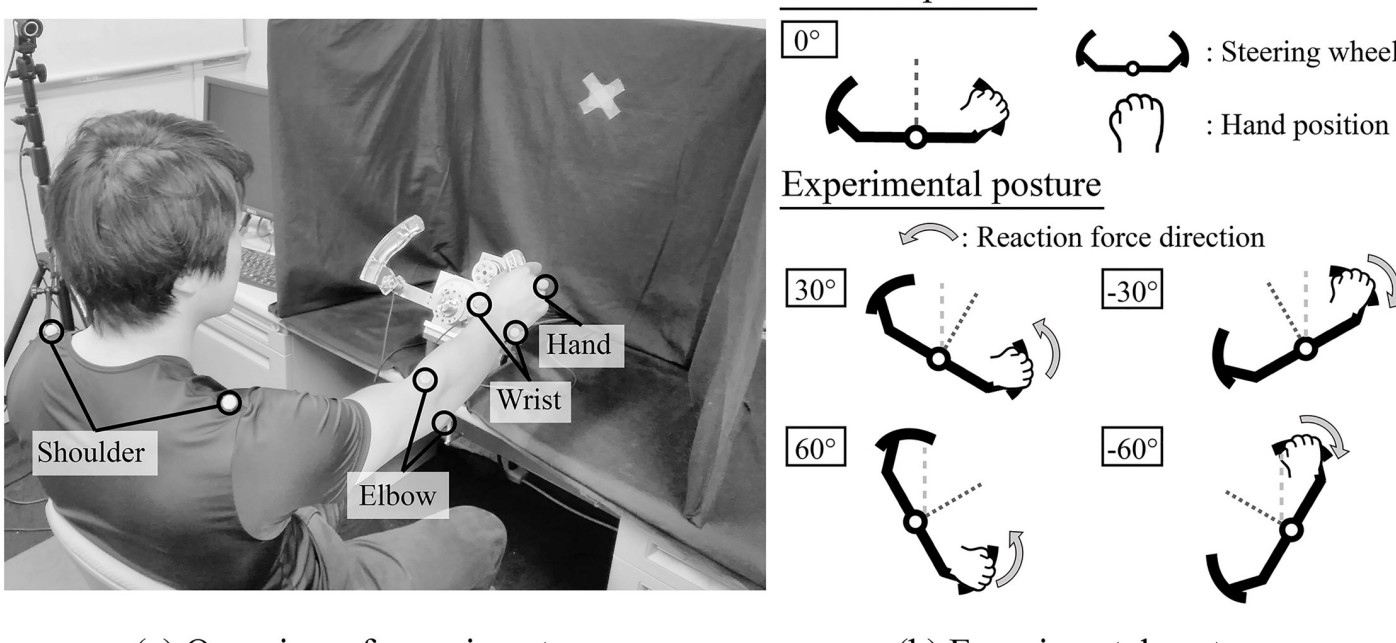

**Fig 2. Experimental conditions.** The subject grasped the right side of the steering wheel device and memorized the reference stimulus in the reference posture(0˚). Afterward, the posture changed to the experimental postures(30˚, 60˚, −30, −60˚) and memorized the test stimuli (1.1–2.9 Nm, staircase method including upward step and downward step). Then, the subject was asked about the larger stimulus. The tasks were repeated 50 times(25 upward step and 25 downward steps) in each experimental posture.

experimental stimuli were changed in ascending or descending stepwise increments of 0.2 Nm between 1.1–2.9 Nm. The experimental postures and magnitude of each test stimulus were decided from the realistic condition [12]. The direction of the force was the same between the reference and the test stimuli. The steering wheel rotated to the left at the experimental postures of 30˚ and 60˚, and the steering wheel rotated to the right at −30˚ and −60˚ because the direction of the steering reaction force was the same as that of the actual steering reaction force. The experimental tasks were as follows.

1. The participant grasped the handgrip with the right hand and memorized the magnitude of the reference stimulus presented in the reference position for 3 seconds. The participant maintained the posture while the stimulus was presented.

2. After changing to the experimental posture, the participant memorized the magnitude of the test stimulus presented for 3 seconds. The participant maintained the experimental posture while the stimulus was presented.

3. The participant was asked which side was larger.

4. The subsequent test stimulus was modified based on the participant's response.

According to the response of each trial, the test stimulus of the next trial for downward step and upward step was changed as follows.

- Answer that the test stimulus was larger than the reference stimulus: reduce the test stimulus by 0.2 Nm.

- Answer that the test stimulus was smaller than the test stimulus: increase the test stimulus by 0.2 Nm.

These procedures were used in both downward and upward step. The downward and upward step were conducted alternately. The test stimulus was repeated at the comparative stimulus of the chance level that is, near the subjective equivalence value. To avoid the effect of fatigue, a break was provided for each posture. To avoid the order effect, the order of the experimental posture was randomized for each participant. A complete experimental session for each participant consisted of 200 steering trials, with 25 upward and 25 downward steps in each posture.

## Muscle activity estimation using a 3D musculoskeletal model

We used OpenSim [22] to calculate the muscle activity in each experimental condition. Muscle strength was calculated using a combination of elastic and contractile elements based on Hill's muscle model reported by Thelen [23]. The muscle parameters, such as the maximum isometric muscle strength $F^M$, optimum muscle fiber length $l^M$, and pennation angle of the muscle, were determined according to a previous study [24]. In the muscle activity calculation, we measured the reference posture ($0°$) and the experimental posture ($30, 60, -30, -60$ ˚) using six motion capture systems (Optitrack, Optitrack Flex3), and joint angle and joint torque were calculated using inverse kinematics and inverse dynamics. The reflex marker was attached to the shoulder, elbow, wrist, and hand, as shown in Fig 2. Muscle strength was determined by optimizing the muscle activity to balance the joint torque. The $m$-th muscle was calculated to satisfy the following Eq 1.

$$\sum_{m=1}^{n} (\alpha_m F_m^0) r_{m,j} = \tau_j. \tag{1}$$

$F_m^0$ is the isometric maximum muscle strength, $\tau_j$ is the joint torque at the $j$-th joint, and $r_{m,j}$ is the moment arm. $\alpha$ represents muscle activity and is a continuous function of $\alpha_m (0 \leq \alpha_m \leq 1)$, which can be regarded as a control signal in the musculoskeletal system [22]. Based on the relationship between the motor unit firing frequency and muscle activity, the higher the motor unit firing frequency, the greater the muscle activity [25]. The moment arm was determined by the $m$-th muscle length $l_m$ and the $j$-th joint angle [26, 27].

$$r_{m,j} = \frac{dl_m}{d\theta_j}. \tag{2}$$

The following shows the relationship between muscle strength $F_m$ and muscle activity $\alpha_m$.

$$F_m = \alpha_m F_m^0 \bar{f}_l(\bar{l}_m) + F_m^0 \bar{F}^{PE}(\bar{l}_m). \tag{3}$$

$\bar{l}_m$ is the normalized muscle fiber length, and $\bar{f}_l(\bar{l}_m)$ is the normalized muscle strength-length relationship. We used the parameter of $\bar{f}_l(\bar{l}_m)$ and $\bar{F}^{PE}(\bar{l}_m)$ from a previous study [24].

## Data analysis

**Perceptual bias.** In the psychophysical experiment, we calculated the perceptual bias to determine whether a perceived force with an experimental posture was perceived differently when compared with a reference posture. The percentage of responses indicating that the test stimulus was "larger" were calculated for each presented comparison. The percentages were

plotted, and a psychometric curve was fitted to the points with a cumulative Gaussian distribution:

$$f(x) = \frac{1}{2}\left(1 + \operatorname{erf}\left(\frac{x - (2.0 + \mu)}{\sigma\sqrt{2}}\right)\right),$$ (4)

where $\mu$ and $\sigma$ are the fitted parameters representing the mean and SD of the curve, respectively. Because some experimental stimuli were presented more often than others, a weighted least squares fit was used [28]. The value of $\mu$ represents the perceptual bias and $2.0 + \mu$ represents the points of subjective equality for a specific session. A positive value represents an overestimate (i.e., the reference stimulus was perceived to be larger than it actually was in the experimental posture). In contrast, a negative value represents an underestimate (i.e., the reference stimulus felt lighter in the experimental posture). The average bias for all subjects was calculated from the experimental results, and the comparison was carried out using Student's t-tests (significance level: 5%) between the reference and experimental postures. Additionally, analysis of variance (ANOVA) was performed between the experimental postures, and pairwise comparisons using the Holm method were performed (significance level: 5%).

**Prediction of perceptual bias from muscle activity.** In the muscle activity estimation, postural data that were obtained using a motion capture system and force data during each trial were used. In operation of the steering wheel, the previous study reported that the arm and shoulder move to make the positive tangential steering force by moving with forward elevation. For the negative tangential steering force, the arm and shoulder move in a downward direction [29]. These movements are created from the deltoid muscle (anterior, medial, and posterior), the pectoralis major muscle (upper and medial portion), the biceps brachii(long and short), and the triceps brachii (long head and lateral part). Therefore, we used these muscles as representative muscles. In this study, we used the average of the above-mentioned four muscles. The muscle activity differences between the experimental postures were compared using ANOVA, and pairwise comparisons were performed using the Holm method (significance level: 5%). In a previous study, we proposed an estimation model of the force perception-change ratio using muscle activity during steering [12]. It is possible to estimate the perceived force using this model. In this model, the magnitude of perceived force was measured using the psychophysical experiment, and logarithmic fitting was performed based on Weber-Fechner's law:

$$F_p = a\log F_a + b.$$ (5)

where $F_p$ is the perceived force, $F_a$ is the applied force, and $a$ and $b$ are coefficients obtained using the least square method. In addition, muscle activity against the force magnitude $F_a$ was estimated using a three-dimensional musculoskeletal model. The muscles used in the muscle activity estimation were the same as those described above. We obtained the linear relationship between the $F_a$ and the muscle activity.

$$\alpha = kF_a + m,$$ (6)

where $\alpha$ is the muscle activity, and $k$ and $m$ are the coefficients obtained using the least-square method. By substituting Eq 6 into Eq 5, we obtained the following equation:

$$F_p = a\log(\frac{\alpha - m}{k}) + b.$$ (7)

Table 1 lists the coefficient values in Eq 7. This equation can be expressed as a function of

**Table 1. Model coefficients.**

| Coefficient | Value |
|---|---|
| $a$ | 11.74 |
| $b$ | −14.91 |
| $k$ | 0.0015 |
| $m$ | 0.00059 |

the muscle activity. Using Eq 7, the perceived force $F_p$ can be predicted from the muscle activity.

We predicted the perceptual bias in each posture using these equations. First, muscle activity was estimated using the stimulus force data and posture (reference and experimental, respectively). The $F_p$ values were then estimated in both conditions, and a comparison was carried out. In cases where the $F_p$ of the test stimulus was larger than that of the reference stimulus, we recorded the response as "larger". The calculation method of the force perceptual bias followed the technique described in the "Perceptual Bias" chapter, and the predicted force perceptual bias $\mu_{predict}$ was calculated. The accuracy was verified by obtaining the correlation coefficient between the true value and the predicted value.

## Results

### Force perceptual bias

In this experiment, we investigated the perceptual bias in driving posture using a psychophysical experiment. Fig 3 shows the results of the psychophysical experiments on the representative subjects. Fig 3(a) shows the trajectory of a given test stimulus during the experiment. It is predicted that the subject overestimated the reference stimulus because the plots are mostly located at positions larger than 2.0. In Fig 3(b), a psychophysical curve was calculated using the results of Fig 3(a). A positive perceptual bias existed because the center of the "larger steering force"(PSE = 0.5) shifted to greater than the reference stimulus. Fig 4 shows the average of the perceptual bias calculated from the results of the psychophysical experiment. The bias for each posture was compared with the reference posture using Student's t-tests (significance level: 5%). Significant differences were found at 30˚ ($t = 2.7$, $p = 0.03$), −30˚ ($t = −9.0$, $p < 0.001$), and −60˚ ($t = −6.5$, $p < 0.001$). No significant differences were observed at 60˚ ($t = 0.16$, $p = 0.9$). An ANOVA revealed significant differences ($F_{1,8} = 28.3$, $p < 0.001$) between each experimental posture. In pairwise comparisons, significant differences were observed at 30˚ versus −30˚($t = 9.4$, $p < 0.001$), 30˚ versus −60˚($t = 7.2$, $p < 0.001$), 60˚ versus −30˚($t = 4.7$, $p = 0.002$), and 60˚ versus −60˚($t = 4.1$, $p = 0.01$). No significant differences were observed at 30˚ versus 60˚($t = 2.6$, $p = 0.06$) and −30˚ versus −60˚($t = 1.3$, $p = 0.2$). The results show that perceptual bias existed in each experimental posture except for 60˚. Additionally, it was shown that there is a significant difference in the size of the bias based on the posture.

### Muscle activity estimation and bias prediction from muscle activity

The psychophysical experiment showed that perceptual biases existed in the experimental postures (except for 60˚). To further investigate the perceptual bias, we estimated the muscle activity in the experimental postures. Fig 5 shows the representative results of the muscle activity estimation. As a representative results, 1.9 was chosen because it was found most

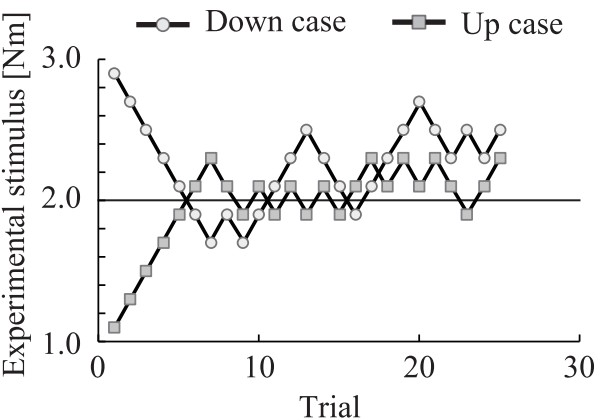

(a) An example of the staircase in 30° of subject A

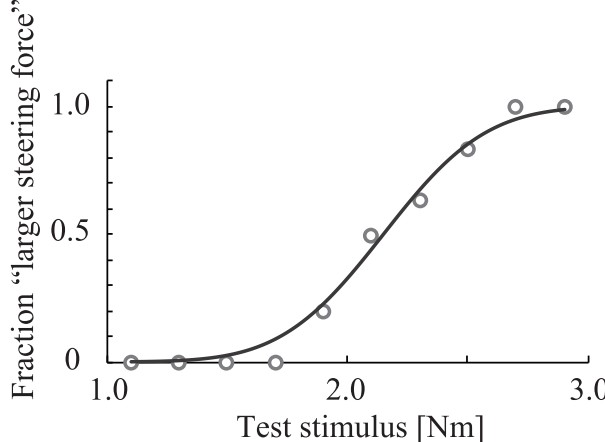

(b) An example of the psychometric curve in 30° of subject A

**Fig 3. A typical result from the psychophysical experiment.** (a) shows the trajectory of a given test stimulus during the experiment on the representative subject. Each step was given 25 times. In (b), each data point shows the percentage of responses in which the test stimulus was reported as "larger", calculated for each presented comparison. The solid line shows the psychometric curve fitted to the answer plot with a cumulative Gaussian distribution using the weighted least-squares method.

frequently among all subjects in the experiment. An ANOVA revealed significant differences ($F = 1.2 * 10^3$, $p < 0.001$) between each experimental posture. In pairwise comparisons, significant differences were observed at 30° versus 60° ($t = -3.35$, $p < 0.001$), 30° versus −30° ($t = -52.2$, $p < 0.001$), 30° versus −60° ($t = -27.1$, $p < 0.001$), 60° versus −30° ($t = -49.7$, $p < 0.001$), 60° versus −60° ($t = -24.3$, $p < 0.001$), and −30° versus −60° ($t = 25.0$, $p < 0.001$). Fig 6 shows the muscle activity estimation result of the reference angle. These muscle activities are compared using Welch's t-tests (significance level: 5%). The result showed a significant difference between the directions of the force($t = -1.93 * 10^2$, $p < 0.001$). Fig 7 shows the plots between the predicted bias $\mu_{predicted}$ and the measured bias. We obtained a significant, moderate correlation ($r = 0.56$, $p = 0.0028$). These results indicate that muscle activity varied with posture, suggesting that muscle activity affected differences in the perceptual bias.

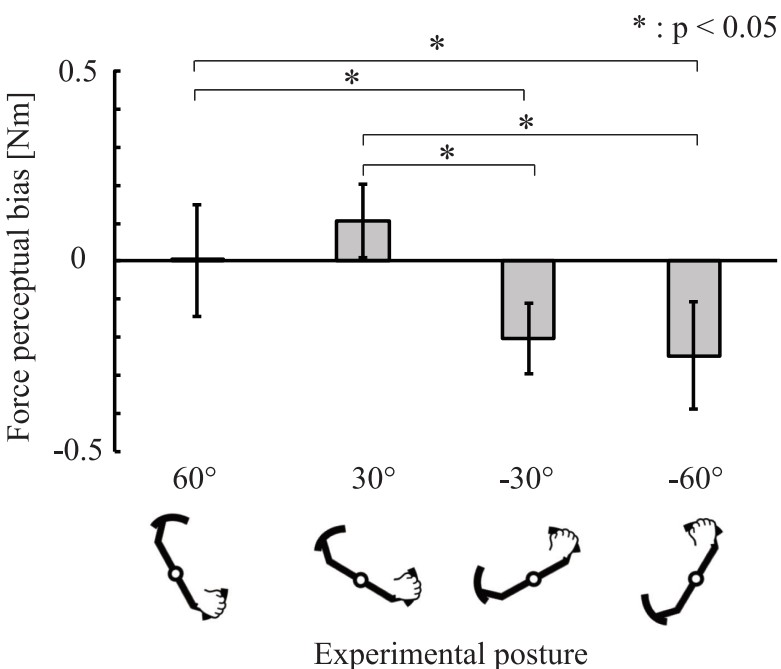

**Fig 4. The result of the psychophysical experiment.** A positive value represents an overestimate (i.e., the reference stimulus was perceived to be larger than it actually was in the experimental posture). A negative value represents an underestimate (i.e., the reference stimulus felt lighter in the experimental posture).

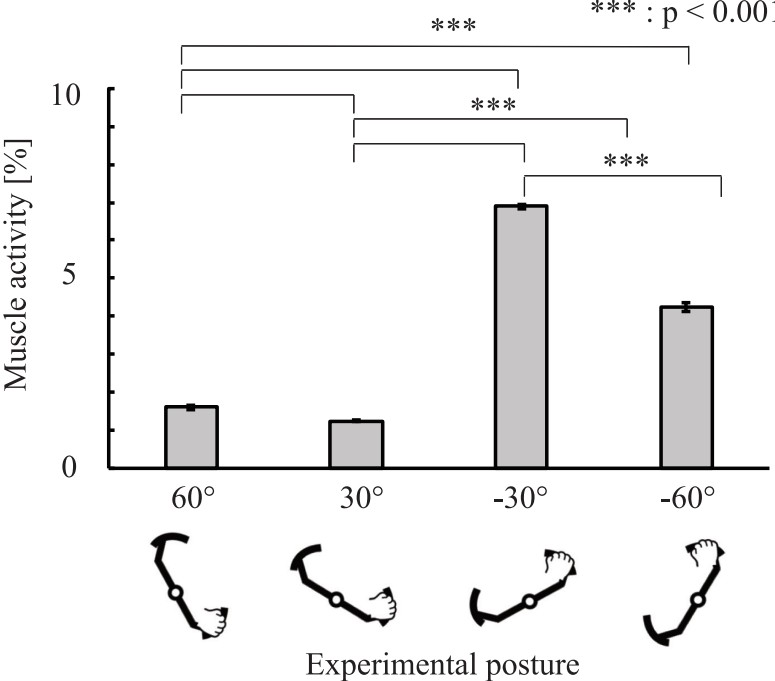

**Fig 5. Mean muscle activity for each angle in all participants (with a stimulus of 1.9 Nm).** The 1.9 Nm test stimulus was used as a force value in the muscle activity estimation because this stimulus trial was the most common across all participants and postures.

*** : p < 0.001

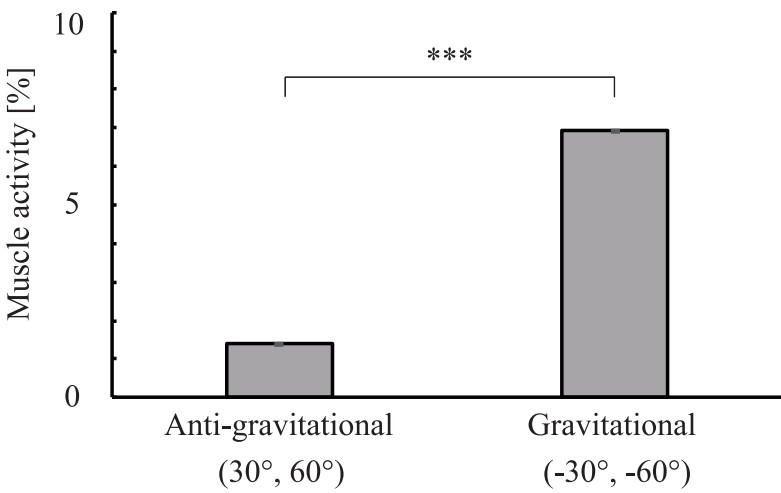

**Fig 6. Mean muscle activity of reference angle in all participants.** In 30˚ and 60˚, the anti-gravitational force is given for the reference force in the 0˚ posture, and the gravitational force is given for the reference force in the 0˚ posture in −30˚ and −60˚. The Welch's t-test showed a significant difference between the directions of the force($t = -1.93 * 10^2$, $p < 0.001$).

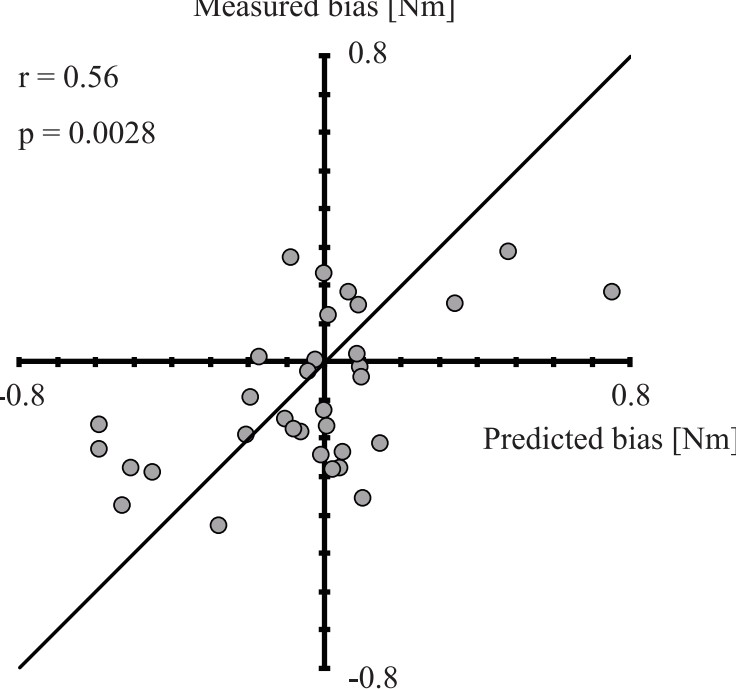

**Fig 7. The scatter plots show the predicted bias from the calculation and the measured bias from the psychophysical experiment.** The solid line is the line of equality, where the predicted bias and measured bias exactly matched. We obtained a significant moderate correlation ($r = 0.56$, $p = 0.0028$).

## Discussion

### Bias and muscle activity

The results revealed significant differences when compared with the reference posture (i.e., force perceptual bias was caused by changing the posture) in all positions except for 60˚. Additionally, the muscle activity estimation was also carried out in each trial. As shown in Fig 5, muscle activity varied depending on the posture, even when the same stimulus was presented to participants. Jones reported that when the weight of an object is discriminated, the relative size is perceived and scaled by the range of muscle activities involved in motion [3, 30]. This finding indicates that high muscle activity could potentially cause perceptual bias.

The results of the psychopsysical experiment revealed a significant difference between postures, as shown in Fig 4. Additionally, significant differences existed between the anti-gravitational and gravitational directions, as shown in Fig 6. These results suggest that the force direction during the trials affected the perceptual characteristics, similar to the results of Sakajiri et al., who reported an effect of whether the force direction was in the gravitational direction or not [15].

Human somatic sensation is known to change depending on whether the direction of the force is in the gravitational direction or not, and many studies have examined the effects of gravity. In the field of developmental psychology, Hood et al. report that infants learn the effect of gravity on objects as they age [31]. People move on the assumption that there is gravity [32], and the weight discrimination threshold rises in zero-gravity space [33]. In addition, Young et al. report that the positional sense of the body is lost, and motor skill decreases, when subjects operate in the absence of vision under zero-gravity space conditions [34]. The direction of the reaction force changes depending on the rotating direction in steering and becomes the anti-gravitational direction depending on the position of the arm. In the 30˚ and 60˚ conditions, the force direction is anti-gravitational because only the right hand gripped the steering wheel in this experiment. The reaction force can be offset by the arm's own weight in the anti-gravitational direction. Therefore, the muscle activity becomes low at 30˚ and 60˚. Perceptual bias would also be expected to be affected by the difference in muscle activity with the direction of force.

### Bias prediction

We conducted a psychophysical simulation experiment to predict bias using estimated muscle activity from postures and arm force data during the experimental tasks. The results revealed a significant moderate correlation between the predicted bias and the actual bias, indicating that human force discrimination could be predicted relatively accurately based on the psychophysical experimental simulation. Since only the estimated muscle activity was used, the prediction made it easier to examine the bias, compared with the experimentally determined muscle activity. Additionally, from the perspective of the force perception mechanisms of the body, it is possible to explain the bias based on muscle activity. Consideration of perceptual bias in steering is useful for designing steering reaction force, and the improvement of operability could play an important role in preventing operational error.

In recent years, however, it has been reported that afferent signals from muscle spindles and skin receptors in the periphery are also important factors in determining sense of force/heaviness [19, 35, 36]. Although it has been confirmed that the sense of effort can be used for judging force/heaviness, an influential current hypothesis predicts that judgments of force/heaviness are based not only on sense of effort but also on feedback of afferent signals

returning from the periphery [37]. Monjo et al. propose that humans do not perceive signals of only efferent or afferent signals as sense of effort but can perceive effort by changes in weight between both signals according to the experimental conditions [38]. The present experiment did not include conditions such as paralysis of muscle spindles. However, as Proske et al. report, it is necessary to provide participants with proper instructions when examining either efferent or afferent signals alone [19]. Since the prediction is carried out only by the muscle activity interpreted as the efferent signal, the afferent signals, such as sensing information from the muscle spindle and cutaneous sensation, which can be considered afferent feedback, appear to affect the prediction accuracy.

Additionally, although the range of steering reaction force is the same in the estimation model, the model was based on psychophysical experiments using both hands. Therefore, the current model cannot be completely applied in this case.

The accuracy of predicting perceptual bias depends on the accuracy of estimating the muscle activity using the musculoskeletal model. In this estimation, muscle co-contraction is neglected in the estimation of muscle activity using our method. Humans are known to perform stable movements by increasing joint stiffness through muscle co-contraction [39–41]. Therefore, it is important to consider muscle co-contraction when estimating muscle activity, to improve estimation accuracy. Additionally, Osu et al. report that muscle co-contraction decreases as humans become accustomed to motor tasks [42]. In other words, it is possible to improve accuracy by reducing the effect of co-contraction by setting experimental conditions under which co-contraction does not occur, or by selecting subjects who are familiar with such motor tasks.

## Conclusion

In the current study, we investigated whether force perceptual bias depends on posture while steering using a psychophysical experiment. The results revealed bias at postural angles of 30˚, −30˚, and −60˚. These findings suggested that muscle activity increases by changing the posture and direction of the reaction force. We predicted the force perceptual bias using muscle activity during the experimental task and obtained a significant moderate correlation between the predicted and measured bias. The results of the prediction indicated that it is possible to predict perceptual bias with relatively high accuracy using muscle activity, interpreted as sense of effort. In future studies, we plan to test steering reaction force conditions considering this perceptual bias, to investigate the relationship with the sensation of steering.

## Author Contributions

**Conceptualization:** Yusuke Kishishita, Yoshihiro Tanaka, Yuichi Kurita.

**Data curation:** Yusuke Kishishita.

**Formal analysis:** Yusuke Kishishita.

**Investigation:** Yusuke Kishishita.

**Methodology:** Yusuke Kishishita, Yoshihiro Tanaka, Yuichi Kurita.

**Project administration:** Yoshihiro Tanaka, Yuichi Kurita.

**Resources:** Yoshihiro Tanaka.

**Software:** Yusuke Kishishita.

**Supervision:** Yoshihiro Tanaka, Yuichi Kurita.

**Validation:** Yusuke Kishishita.

**Visualization:** Yusuke Kishishita.

**Writing – original draft:** Yusuke Kishishita, Yoshihiro Tanaka, Yuichi Kurita.

**Writing – review & editing:** Yusuke Kishishita, Yoshihiro Tanaka, Yuichi Kurita.

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
