## [Decision Letter · Decision Letter 0]

24 Jul 2019

PONE-D-19-16009

Force perceptual bias caused by muscle activity in unimanual steering

PLOS ONE

Dear Mr. Kishishita,

Thank you for submitting your manuscript to PLOS ONE, and I'd like to personally apologize for the delay in handling the manuscript. 

After careful consideration, we feel that it has merit but does not fully meet PLOS ONE’s publication criteria (https://journals.plos.org/plosone/s/criteria-for-publication) as it currently stands. Therefore, we invite you to submit a revised version of the manuscript that addresses the points raised during the review process.

Both reviewers and I agree that more information, and possibly new analyses, are needed to comply with Criterion #4 (Experiments, statistics, and other analyses are performed to a high technical standard and are described in sufficient detail). Please see Reviewer 1's comments regarding experimental details that seem to be missing. Also, some analyses and model equations may not be appropriate, as pointed out by Reviewer 2. I too wonder whether the conclusions would remain the same if torque direction were treated separately from posture per se, whether averaging muscle activity (e.g. across agonist/antagonist) was appropriate, and why EMG measurements were not made.

Additionally, to ensure adequate compliance with Criterion #3 (Conclusions are presented in an appropriate fashion and are supported by the data), please address the comments from both reviewers about the clarity of hypotheses, predictions, and conclusions. The manuscript is admirably concise, but sometimes too much so, leaving the reader with important unanswered questions. 

Lastly, it was not entirely clear whether data will be uploaded as part of Supplementary Information, or uploaded to a public repository, as outlined in Criterion #7.

We would appreciate receiving your revised manuscript by Sep 07 2019 11:59PM. To enhance the reproducibility of your results, we recommend that if applicable you deposit your laboratory protocols in protocols.io, where a protocol can be assigned its own identifier (DOI) such that it can be cited independently in the future. For instructions see: http://journals.plos.org/plosone/s/submission-guidelines#loc-laboratory-protocols

We look forward to receiving your revised manuscript.

Kind regards,

Christopher R. Fetsch

Academic Editor

PLOS ONE

Journal Requirements:

This work was partially supported JSPS KAKENHI Grant Number JP17J06986, JST, PRESTO Grant Number JPMJPR16D3 and laboratory exchange program via Special Interest Group for Haptics, The Virtual Reality Society of Japan.

JSPS KAKENHI Grant Number JP17J06986

JST, PRESTO Grant Number JPMJPR16D3

https://ieeexplore.ieee.org/document/8341739

In your revision ensure you cite all your sources (including your own works), and quote or rephrase any duplicated text outside the methods section. Further consideration is dependent on these concerns being addressed.

Reviewers' comments:

Reviewer's Responses to Questions

**Comments to the Author**

1. Is the manuscript technically sound, and do the data support the conclusions?

Reviewer #1: No

Reviewer #2: Partly

2. Has the statistical analysis been performed appropriately and rigorously? 

Reviewer #1: I Don't Know

Reviewer #2: No

3. Have the authors made all data underlying the findings in their manuscript fully available?

Reviewer #1: No

Reviewer #2: Yes

4. Is the manuscript presented in an intelligible fashion and written in standard English?

Reviewer #1: Yes

Reviewer #2: Yes

5. Review Comments to the Author

Reviewer #1: The framework of this study is difficult to understand, as the introduction does not provide a clear link between sense of effort and posture. Further, there is no clear statement of hypotheses/predicted outcome, which, given the limited framework of the introduction make it difficult to interpret the purpose of the study. Further, there is no description of the raw data that was collected and only report of a reduced bias term in a very limited results section – this does not give the reader much to work with in terms of interpreting the results of the study. There are many details missing from the methods, including how trials were counterbalanced, other than report that they were not equal. This is worrisome, as having more “heavy” trials can bias subjects to make judgements that a stimuli is more “heavy” and vice versa. A properly counterbalanced study will allow for careful examination of perception without the confound of unequal trial presentation (and a more "true" judgement of whether or not something requires more perceptual effort). Additionally, it appears that they are adopting parameter values in the muscular model that were originally described in studies of 30-70 year old subjects, while this population is ~20 years old.

Major

1) The abstract is difficult to read and needs to clearly portray the motivation of the study.

2) The Introduction is lacking many details to give the reader the necessary information to support the motivation for the study: Line 18-19 – what are the physical and psychological factors impacting sense of effort? What do the numbers reported for Stevens’ power law mean about actual perceived and executed muscular effort? Why isn’t there a description of Weber-Fechner’s law (some readers likely do not know what this is)? How does muscle activity change the sense of force? It would help the reader to have some insight into the physiologic basis of WHY perception of effort occurs and why it is important, especially since their study collects and models EMG data. Overall, the introduction just gives a list of studies, but does not provide any framework with which to understand the motivation of the authors experiments.

3) I would recommend refraining from referring to upper limb control as posture, as most readers in the field of motor control would assume that it refers to lower limb posture and balance.

4) Line 57-59: The authors do not present sufficient evidence that “sense of force and heaviness can be estimated from muscular activity, which can be interpreted as sense of effort”- it is unclear how this statement is true given the evidence that the authors have presented.

5) The authors do not present any information for the specifics of the staircase methodology – what specific steps were used? What happened if subjects made an error? Why were the postures of 30 and 60 degrees chosen? Line 11 – How were experimental stimuli modified based on the participant’s response?

6) This task has a significant memory component, in that subjects have to hold the posture for 3 seconds and memorize it, how do the authors account for aspects of memory that may contribute to the task?

7) Line 3 - Heaviness typically refers to the weight of an object (eg., holding it and testing weight in your hand). The subjects are not supporting the steering wheel, but are instead gripping the wheel in response to a force being applied. It is unclear whether the authors are testing response to heaviness or response to an applied force. As the directions they have supplied to subjects do not match up with the description of the research in the introduction.

8) Line 114-116: The authors describe 200 trials, with 25 repetitions in each mode, but there is no description of what these modes were. Clarification and explicit statements are experimental procedures are needed here.

9) Why wasn’t EMG collected and then confirmed via the model-based approach?

10) The utilization of parameters proposed by Thelen is surprising given that this original citation examined age-related changes in biomechanics from age 30-70 (young vs old adults). The sample here is roughly ten years younger and it is likely that these same parameters do not apply to younger adults.

11) The results section of this manuscript is two small paragraphs. This gives the reader very little information to go on in terms of what the outcome of this study is and how it relates to their hypotheses and framework.

Minor

1) Perceptual bias needs to be clearly defined in the abstract

2) 9 subjects seems like far too few for a study like this. Is this study appropriately powered? Why was one of the subjects unable to drive? Does this subject have extensive driving experience?

3) Line 113 – How long were the breaks that subjects were allowed?

4) Why weren’t the experimental stimuli evenly counterbalance. It is possible that presenting with more “heavier” trials will bias subjects to over-estimate, based on previous history of stimuli.

5) Figure legends need more description.

Reviewer #2: The manuscript describes a study using a steering wheel where different forces (torques) are applied on the hand when holding the wheel in different orientations. The authors show that the orientation results in biases of the perceived force. Furthermore, they link this to differences in muscle activations that were calculated from the measured reaction forces using a muscle model.

I think this is an interesting study, relating muscle activity to force perception. However, I have some concerns about the terminology and the data analysis/methods. Although the results are nice, I think some of the analyses are not valid for the current data set and this affects the interpretations. Also, I believe the results as they are now do not only show effects of posture, but also of torque direction. The authors should perform new analyses to address these comments.

1) First of all, the authors report that they investigate effects of force. However, since the task concerns a rotation of the steering wheel, this is actually about torques, not forces. Indeed, the biases are reported in Nm, not N. Although the participants are asked to report which stimulus was heavier, the authors should clarify that they measure torques.

2) One main concern about the methods is that the different postures also involve different torque directions. I see that this was done to have realistic conditions in driving and steering, but this could influence the results. In fact, it seems that the different orientations do not differ in perceptual bias (i.e. no difference between 30 and 60 degrees, neither between -30 and -60 degrees), but the bias direction (i.e. negative or positive) is different. So the biases also seem to be driven by the direction of the torques. By altering the direction, the authors seem not to have not only investigated the effect of posture (with different muscle contributions) on force perception, but also the effect of torque direction (which should also lead to different muscle contributions based on the direction of the force that is applied by the arm) on perception. Since the same torque direction was used in the reference as the experimental condition, this still investigates an effect of posture for each condition on its own. Therefore, I think the results are still interesting, since significant biases were found, but only postures with the same force direction can be compared. In sum, I think the authors should split their analysis by the two torque directions.

3) Related to this, I think the authors should report the muscle activity for the reference condition as well. These were performed with two different torque directions. If the muscle activity differs between these directions in the same posture, this would already show that the muscle activity also varies because of torque direction effects, not only posture.

4) Another concern I have about the methods is the muscle model. To me, it seems strange that the muscle activity of agonist and antagonists are averaged. They mention this as a limitation in the discussion, but I think the muscle model as it is now is not very insightful. The average activity of multiple muscles can give similar values for very different combinations of different muscles. Perhaps they could calculate the activity for different muscles (or at least only average over agonist and antagonist muscles). If the activity pattern is similar in different tasks and only increases overall, this would validate their decision to average the muscle activity, but now this is unknown.

5) It seems to me that the calculations for the predicted bias are unnecessarily complicated. I do not think the perception-change ratio is needed to do this. When equation 5 and 7 are combined: Fp=a log (alpha-m/k)+b. Therefore, equation 6, 8 and 9 do not seem necessary.

6) I do not understand where the values in Table 1 come from. Are they from a previous paper or are they fitted to the data? If the latter is true, what data is exactly compared to which model to obtain these parameter values?

7) The authors correlate the predicted biases from the muscle activity/measured forces to the measured biases. However, I wonder whether they can do it the way they have done it now. First, similarly to my earlier comment about comparisons between the torque directions, it might not be fair to correlate over two different torque directions. The biases in one direction are negative, while the biases in the other direction are positive. I am not sure whether it is justified to combine these two conditions into a single plot. Secondly and related to this, the authors seem to have combined different data points (for different conditions) for the same subjects into one plot. Different conditions for one subject are within factors, whereas data points for different subjects have between factors and therefore have different sources of variation. I am not an expert on these statistics, but I wonder whether this can be combined in a single correlation. Perhaps a multilinear regression model might be more suitable.

8) Do the authors have an explanation why they do not find a bias for the 60 degree condition? This does not follow from the muscle activity pattern, so perhaps the authors could indicate other factors that drive the perception?

Minor comments:

a) P2 L46 I am not sure whether visual information can be considered a cognitive feature. This could also be more low-level sensory effects that contribute to the ultimate heaviness perception.

b) The authors report that they counterbalance the conditions to avoid order effects. However, it seems that the reference condition was always performed before the experimental condition. Could this not also lead to order effects or biases? See for example the time-order error as described by Hellstrom (1985, Psychological Bulletin; 2003 Perception & Psychophysics).

c) P5 L158 The authors state that they test whether the bias was different between the standard and experimental procedure. However, from the psychometrical curve, the bias is defined as the difference between the standard and the experimental stimulus, so there can be no bias calculated for the standard position. Do the authors mean they tested whether the biases were significantly different from zero?

d) P6 L195 I think this should be Eq. 8, not Eq. 1?

e) The author contributions are not described (P 9).

Textual comments:

- P1 L18 "..to sense of force..." remove "of"

- P1 L24 "Thus" seems strange here, since fatigue affects do not relate to the previous sentence about colour affects. Perhaps replace with "In addition".

- P5 L178 "ANOVA was performed" -> An ANOVA was performed

6. PLOS authors have the option to publish the peer review history of their article (what does this mean?). If published, this will include your full peer review and any attached files.

Reviewer #1: Yes: Jennifer Semrau

Reviewer #2: No

---

## [Author Response · Author response to Decision Letter 0]

13 Sep 2019

Thank you for your valuable comments.

We have revised the manuscript according to the comments from editor and reviewers.

All answer are including in attached files.

Thank you in advance.

Respectfully,

Yusuke Kishishita

---

## [Editor Report · Decision Letter 1]

24 Sep 2019

PONE-D-19-16009R1

Force perceptual bias caused by muscle activity in unimanual steering

PLOS ONE

Dear Mr. Kishishita,

Thank you for submitting your manuscript to PLOS ONE. After careful consideration, we feel that it has merit but does not fully meet PLOS ONE’s publication criteria as it currently stands. Therefore, we invite you to submit a revised version of the manuscript that addresses the points raised by the Academic Editor. (A point-by-point rebuttal letter is not necessary for this revision, as there are no additional comments by the Reviewers.)

Publication Criterion #5 states: “The article is presented in an intelligible fashion and is written in standard English.” Now that the scientific concerns raised by the reviewers have been addressed, it was easier to discern whether this criterion has been met, and the short answer is no, it has not. The manuscript requires additional proofreading and revision to improve clarity, preferably with the help of an expert in written English. [Fortunately I have done much of this work for you, see below. -CF] We apologize for not raising this concern in the previous round, but it was more critical to get the scientific issues out of the way first.

Note that PLOS ONE does not copyedit manuscripts, and thus what may seem like minor errors that would be handled by editorial staff at other journals must here be rectified by the authors. In general, things to look out for include: missing articles (a, the), verb tense and subject-verb agreement, and pluralization of nouns, and spelling errors. I had planned on highlighting just two or three examples, but in the process I ended up writing down every error I found, so I might as well provide you the whole list (of course there may be others that I missed):

line 12: acitivity -> activity.

12-14: The two sentences “This study enable us…” and “In addition, it can…” are difficult to understand, and seem to relate more to broader implications or extensions of the study rather than the study itself. I would omit them, or move them to Discussion.

38: “These power represent”

Fig 2 Legend: gripped, not griped. Memorized, not memorize. Were, not are.

101-113: several singular/plural errors etc. in this section

132: “the test stimulus is smaller than the test stimulus” — you mean reference?

147: “according to a previously investigated [23].”

189: “In operation of steering wheel, we mostly used the arm and shoulder to make positive tangential steering force by moving forward elevation” — This sentence is problematic because “we…used” implies you are referring to the experimenter, not drivers in general, and thus the reader is expecting the sentence to describe something about the task or apparatus rather than a justification for including particular muscle groups in the muscle activity analysis.

197: “, as the average of the four muscle activities described above” repeats an earlier part of the sentence and can be omitted.

197-199: Sentence about ANOVA+Holm is nearly identical to the one on 184-186.\\

228: semicolon or period instead of comma

231: larger steering force, not larger steering wheel

Fig 3b legend: each data point, not “each plot”

Legends for Figs 4+5: the statistics don’t need to be repeated verbatim from the results text.

238 & 253: “In multiple comparison,“ recommend using the word “pairwise”, e.g. “In [multiple] pairwise comparisons”, and also suggest using “vs.” or “versus” instead of an dash (in for example  “30 — -60”) so it’s easier to read when the angles are negative.

247: “showed that the perceptual bias exist” —> showed that perceptual biases exist, or bias exists

256: “ To evaluate the muscle activity in different direction of force,”  ??

257: “muscle activity estimation result of reference angle” ??? As a result of?

274: “of psychopsysical experiment” — misspelled and missing a “the”

275: “ the significant difference exist”

I found the rest of this subsection of Discussion to read pretty well, so I stopped proofreading at that point. However you may want to check over the remaining sections and double-check the figure legends and axis labels etc. as well. 

We would appreciate receiving your revised manuscript by Nov 08 2019 11:59PM. To enhance the reproducibility of your results, we recommend that if applicable you deposit your laboratory protocols in protocols.io, where a protocol can be assigned its own identifier (DOI) such that it can be cited independently in the future. For instructions see: http://journals.plos.org/plosone/s/submission-guidelines#loc-laboratory-protocols

A marked-up copy of your manuscript that highlights changes made to the original version. This file should be uploaded as separate file and labeled 'Revised Manuscript with Track Changes'.An unmarked version of your revised paper without tracked changes. This file should be uploaded as separate file and labeled 'Manuscript'A point-by-point rebuttal letter is not necessary for this revision.

We look forward to receiving your revised manuscript.

Kind regards,

Christopher R. Fetsch

Academic Editor

PLOS ONE

---

## [Author Response · Author response to Decision Letter 1]

30 Sep 2019

Thank you for reviewing my language issue and sorry for bothering your time.

We edited the manuscript and submitted the language editing service.

Please see our manuscript again.

Thank you in advance.

Respectfully,

Yusuke Kishishita

---

## [Editor Report · Decision Letter 2]

2 Oct 2019

Force perceptual bias caused by muscle activity in unimanual steering

PONE-D-19-16009R2

Dear Dr. Kishishita,

We are pleased to inform you that your manuscript has been judged scientifically suitable for publication and will be formally accepted for publication once it complies with all outstanding technical requirements.

With kind regards,

Christopher R. Fetsch

Academic Editor

PLOS ONE
---

## [Editor Report · Acceptance letter]

10 Oct 2019

PONE-D-19-16009R2 

Force perceptual bias caused by muscle activity in unimanual steering 

Dear Dr. Kishishita:

I am pleased to inform you that your manuscript has been deemed suitable for publication in PLOS ONE. Congratulations! Your manuscript is now with our production department. 

With kind regards,

on behalf of

Dr. Christopher R. Fetsch 

Academic Editor

PLOS ONE